# The Novel Diagnostic Techniques and Biomarkers of Canine Mammary Tumors

**DOI:** 10.3390/vetsci9100526

**Published:** 2022-09-26

**Authors:** Ilona Kaszak, Olga Witkowska-Piłaszewicz, Kinga Domrazek, Piotr Jurka

**Affiliations:** 1Laboratory of Small Animal Reproduction, Department of Small Animal Diseases and Clinic, Institute of Veterinary Medicine, Warsaw University of Life Sciences, 02-787 Warsaw, Poland; 2Department of Pathology and Veterinary Diagnostics, Institute of Veterinary Medicine, Warsaw University of Life Sciences, 02-787 Warsaw, Poland

**Keywords:** canine mammary tumors, biomarkers, dog, breast cancer, bitch, neoplasia

## Abstract

**Simple Summary:**

The high rate of cancers in both humans and in animals along with their poor overall survival pose a major challenge in medicine. Therefore, a large amount of research has been focused on improving the diagnosis and efficacy of treatment. In the case of human breast cancer (HBC) there are several diagnostic and treatment protocols, which greatly decrease the rate of death among patients. In veterinary medicine, canine mammary tumors (CMTs), which are very similar to HBC, are still poorly studied, lacking in treatment options, they are diagnosed at an advanced stage, and usually have a bad prognosis. Therefore, early diagnosis of CMTs and introducing personalized treatment are necessary. This article is a summary of possible novel diagnostic techniques as well as the main biomarkers that seem to improve treating CMTs.

**Abstract:**

Canine mammary tumors (CMTs) are considered a serious clinical problem in older bitches. Due to the high malignancy rate and poor prognosis, an early diagnosis is essential. This article is a summary of novel diagnostic techniques as well as the main biomarkers of CMTs. So far, CMTs are detected only when changes in mammary glands are clinically visible and surgical removal of the mass is the only recommended treatment. Proper diagnostics of CMT is especially important as they represent a very diverse group of tumors and therefore different treatment approaches may be required. Recently, new diagnostic options appeared, like a new cytological grading system of CMTs or B-mode ultrasound, the Doppler technique, contrast-enhanced ultrasound, and real-time elastography, which may be useful in pre-surgical evaluation. However, in order to detect malignancies before macroscopic changes are visible, evaluation of serum and tissue biomarkers should be considered. Among them, we distinguish markers of the cell cycle, proliferation, apoptosis, metastatic potential and prognosis, hormone receptors, inflammatory and more recent: metabolomic, gene expression, miRNA, and transcriptome sequencing markers. The use of a couple of the above-mentioned markers together seems to be the most useful for the early diagnosis of neoplastic diseases as well as to evaluate response to treatment, presence of tumor progression, or further prognosis. Molecular aspects of tumors seem to be crucial for proper understanding of tumorigenesis and the application of individual treatment options.

## 1. Introduction

In recent years, in both humans and animals the rate of cancers has greatly increased [1,2]. Human breast cancer (HBC) is the second most common tumor to be diagnosed in women. Unfortunately, HBCs are usually malignant. The evaluation of the malignancy relies on the type of tumor, presence of significant nuclear and cellular pleomorphism, mitotic index, presence of necrotic areas, peritumoral and lymphatic invasion, and regional lymph node metastases [3]. Fortunately, the increased number of studies have introduced advances in the diagnosis and treatment of HBCs. 

The great similarity between canine mammary tumors (CMTs) and HBCs in their biology and physiopathology together with the risk of disease recurrence and presence of metastasis to other organs, for example, the lungs or liver, leads to the possibility of performing comparative studies between those species [4,5]. Due to these similarities in histological structure as well as in molecular features, CMTs are regarded as a model for HBCs studies [4,6,7]. The CMTs are the most frequently diagnosed tumors in non-spayed bitches [8,9,10,11]. Approximately 50–70% of CMTs are malignant [9,12]. The most common types of CMTs are adenocarcinoma; papillary carcinoma, solid carcinoma, complex carcinoma, and carcinosarcoma [9,10]. The non-malignant mammary tumors are mostly fibroadenomas, ductal papillomas, benign mixed tumors and simple adenomas [9,10]. Frequently, two or more tumor types are found in various mammary glands of the same dog. On the other hand, in women the most frequent type of HBCs is invasive ductal carcinoma (NOS) [13].

The etiology of CMTs is still unknown, but important risk factors like hormonal, nutritional, and genetic factors are described [14,15,16]. Sex hormones are among the major risk factors for tumor development [14,15,16]. So far, the therapeutic gold standard method is the removal of the tumor, adherent glands, and local lymph nodes. Most surgeons at the same moment perform an ovariohysterectomy (OHE). However, some investigations showed that OHE is not beneficial for every case of CMTs [17]. 

The prognostic assessment both in humans and dogs is built on the patient’s clinical evaluation, tumor size, macroscopic evidence of metastases on the thorax radiographs, lymphatic or vascular invasion, and results of the histopathological examination of the tumor [1,8,14,18,19,20,21]. However, the early diagnosis of HBCs and CMTs is significant for the patient’s outcome due to the high malignancy rate. Proper diagnostics of CMT is especially important as they represent a very diverse group of tumors and therefore different treatment approaches may be required. For this purpose, the specific biomarkers which could be assessed in serum and/or tumor tissue are used. This measurement may not only confirm the neoplastic process but also help to evaluate courses of treatment and prognosis [22,23]. In human medicine, biomarkers are important in HBCs diagnostic. Many biomarkers turned out to be validated in both CMTs and HBCs diagnoses.

This article reviews the most novel diagnostic techniques and biofluids or tissue biomarkers of the CMTs.

## 2. Diagnostic Techniques

### 2.1. Cytology and Histological Grade of Canine mammary Tumors 

As the constantly aggravating risk and augmenting deaths caused by HBCs continues to be a grave concern, the past few decades of cancer research have propelled toward the search for a model organism that can mimic and provide a better understanding of the underlying molecular pathology behind HBCs [4]. Cytological examination (cytology, cytopathology) offers several advantages as it is simple, cost-effective, easy to use, relatively non-invasive, and quick in providing results [24]. In humans, Fine Needle Aspiration Cytology (FNAC) has a standard protocol; known as the Robinson’s grading system, which is highly correlated with the established Scarff Bloom-Richardson histological grading system [4]. FNAC is commonly applied for cytological differentiation of CMTs and analysis of their different cellular origins [25], but the use of FNAC for cytological grading is not clear [4]. 

The sensitivity and specificity of cytological evaluation of CMTs compared to histopathological examination are 65–88% and 94–96%, respectively [26]. The cytological grading system of HBCs can also be used for the diagnosis and prognosis of CMTs. Although the accuracy is about 88.5% [27], this technique is not applied for the CMTs grading [4]. Recently, the use of different cytological techniques in veterinary medicine gained popularity [24]. However, the final diagnosis of CMTs is usually based on the histopathological evaluation of the tumor sample [4]. The literature overview reveals that there is only one report demonstrating a cytological grading system (CGS) in dogs by FNAC [24] and one review [4] in which a new CGS was proposed based on the literature. Dolka’s group proposal was a modification of Robinson’s grading system of HBC. The low sensitivity and specificity of CMTs cytological examination in comparison to human studies might be due to CMTs heterogeneous morphology [24]. This observation requires adequate sample collection and a cytopathologist with experience in the CMTs evaluation [4]. According to Dolka’s team, cellularity and cytological background had no impact on the patient’s life expectancy, whereas, in the case of Grade 2 and Grade 3 CMTs, the period of the patient’s survival was shorter. According to Kuppusamy’s team, cellular dissociation in cytological specimens could not play a role as a prognostic factor for nodal metastasis of CMTs, in contrast to the study on HBCs [28]. The new CGS of Kuppusamy’s team based on literature needs confirmation on a scientific basis. However, it seems that establishing a reliable and accurate cytological classification system in CMTs still requires a collective effort by researchers.

The classical staging system of mammary carcinomas (SSMC) of bitches has an influence on prognosis and further therapy, among other chemotherapy. SSMC relies on the tumor size (T), nodal stage (N), and distant metastasis (M). However, the TNM staging system of CMTs is not very precise, as the examination of the nodal stage is made by palpation, cytology, and/or diagnostic imaging [29]. The tumor size measured clinically by caliper can be overestimated as it includes skin and adipose tissue thickness, and also mammary hyperplasia adjacent to the neoplasia [30]. In the case of HBCs stages I–III, a more accurate and valid staging system is applied, in which evaluation is done by the pathologist, and changes include pathologic tumor size (pT) and pathologic nodal stage (pN) [30]. Actually, in HBCs 2 classification systems (the World Health Organization [WHO] classification from 1999 and the proposal in 2011) and two grading methods based on the human Nottingham grade are used [30,31]. Ana Canadas and her team simultaneously applied the 2 classifications to 2 grading methods for CMTs. In both systems, the histological subtype significantly correlated with the survival rate. In the actual histological staging system [30] a specific category for CMTs mammary carcinomas in situ (stage 0) exists, like in HBCs: stage 0 breast cancers are pTis (carcinoma in situ), N0, M0 [32]. The update from 2017 does not change the pT and pN categories and the anatomical stages of breast cancer but includes pT, pN, M, ER, PR, HER2, and the combined histological grade for patients who received surgery as initial treatment [30,33,34]. However, the histological subtype and grade are considered prognostic factors, a comprehensive comparative study of CMTs classification and grading systems is still missing [31].

### 2.2. Evaluation of Selected Methods of Ultrasound Examination of Mammary Tumors in Bitches

The clinical staging in CMTs involves the evaluation of tumors and sentinel lymph nodes that drain these tumors. Since non-invasive diagnosis is an important requirement in both human and veterinary medicine, simple and available ultrasound techniques like a B-mode ultrasound, Doppler technique, contrast-enhanced ultrasound (CEUS), and real-time elastography are more and more popular. In most publications on non-invasive diagnosis and differentiation between benign and malignant tumors of the mammary gland in bitches, the use of algorithms with various ultrasound techniques is recommended. These procedures increase the sensitivity and efficiency of diagnosis. Soler et al. 2016 researched volume, margins, presence of a capsule, echotexture, presence and distribution of the vascular flow of the tumors by B-Mode, color, power as well as spectra Doppler. The echotexture and type of vascular flow pattern of CMTs support the initial differentiation between benign and malignant tumors. However, the final diagnosis is based on the histological study [35].

A different algorithm in foretelling tumor malignancy in 300 canine mammary masses, using B-mode, Doppler, CEUS, and Acoustic Radiation Force Impulse (ARFI) elastography and compared with histopathology results was evaluated [36]. In short, B-mode and Doppler ultrasound studies have moderate sensitivity and specificity in the malignancy prediction of CMTs. However, the ARFI elastography shear velocity (SWV > 2.57 m/s) with a sensitivity of 94.7% and specificity of 97.2% was an accurate predictor. Thus, the inclusion of ARFI elastography examination in veterinary clinical oncology and research is suggested, as it allows a fast, non-invasive, and complication-free malignancy prognosis of CMTs [36]. In another study, the diagnostic accuracy of B-mode, Doppler, CEUS, and ARFI in identifying a high degree of CMTs malignancy prior to their histopathological classification, including type (simple, complex, or special) and grade (I, II, or III) were evaluated [37]. However, in this study, the accuracy of these ultrasound techniques used for the identification of some of the characteristics of high-grade mammary carcinoma types and grades in bitches was limited (sensitivity from 62 to 68% and specificity from 60 to 62%) [37]. These facts are also confirmed by Vannozi et al. (2018) studies in which identification of ultrasound criteria that may contribute to differentiating between benign and malignant lesions of small size (diameter smaller than 2 cm) was evaluated. However, it was concluded that standard ultrasound examination in bitches in case of small lesions, had a restricted capability of differencing benign from malignant CMTs.

In a recent study, sonoelastography (SE) was confirmed to be more effective than color Doppler US for differentiating malignant from non-malignant CMTs [38]. In this study, the moderate sensitivity and specificity for B-mode (97.0%; 77.9%), SE score (93.9%; 87.0%), color Doppler (81.8%; 66.2%), elasticity to B-mode (E/B ratio) (72.7%; 77.6%) and strain ratio (SR) (77.3%; 79.6%) were obtained.

The status of sentinel lymph nodes (SLNs) is essential in staging, treatment, and further prognosis. Examining the status of sentinel lymph nodes with the elementary and common ultrasonographic techniques can be convenient for staging malignant CMTs in bitches with high accuracy in detecting metastases in SLNs (92.2%) [39]. It was confirmed in the latest study where an ultrasonographic examination that included B-mode, Doppler technique, CEUS, and elastography was compared to a histopathological examination [39]. The best precision in identifying metastases in SLNs was achieved by the elasticity score followed by the short/long-axis ratio and resistivity index, whereas CEUS was as accurate as Doppler.

In order to increase the effectiveness of non-invasive clinical diagnostics, a very interesting direction of research is the combination of ultrasound examination algorithms with the evaluation of selected cancer biomarkers. In recent years, strain ratio (SR) combined with molecular and serum biomarkers for the diagnosis of HBCs were evaluated [40]. It occurred that SR is a very specific and sensitive technique for distinguishing benign from malignant CMTs small lesions (≤1.5 cm). In addition, SR combined with CA15-3 and CK5/6 showed 94.2% sensitivity and 89.2% specificity in comparison to triple-negative (TN) mammary cancer biomarkers, whereas SR combined with D2-40 and CK19 occurred to be reliable diagnostic markers for HBCs lymph node metastasis [40].

Systematic randomized clinical trials in dogs with CMTs are needed in order to improve therapy outcomes and prognosis. Following the One Health concept, human and veterinary medicine should share and combine economic and technological resources of the HBC research with the studies of CMTs to obtain more precise and significant results.

The next sections will describe different groups of serum and tissue biomarkers in CMTs. In order to organize all the information regarding the diagnostic techniques we propose an algorithm of CMTs diagnosis which we use in our practice (Figure 1).

## 3. Biomarkers of Canine Mammary Tumors

### 3.1. Various Cell Markers of the Cancer Process

#### 3.1.1. Cell Cycle Markers

The cell cycle and its regulation are necessary for many processes, such as embryogenesis, growth, reproduction, proper function of adult tissues, and tumorigenesis [41,42]. Overexpression of growth factors or lack of suppressor proteins can be a reason for fast, uncontrolled cell division which allows tumor appearance [42,43]. The cyclins and cyclin-dependent kinases (CDK) play a main role in cell cycle regulation by creating complexes and catalyzing progression through the cell cycle when activated [42]. The family of Cyclins D consists of 3 D-type cyclins: D1, D2, and D3 are associated with two types of CDK’s: CDK4 and CDK6. Overexpression of these cyclins results in rapid cell growth, bypass of key cellular checkpoints, and a neoplastic process [44]. The cyclin D1 overexpression is commonly related with metastasis and shorter life expectancy in HBCs patients [45]. In addition, there is a positive correlation between overexpression of cyclin D1 and overall survival, small size of the primary tumor, low histological grade, and positive estrogen receptor (ER) status [46]. Also, in CMTs the overexpression of the cyclin D1 occurs in 60% of the pre-neoplastic lesions and 44% of neoplastic lesions [47]. However, in another study, the expression of the D1 cyclin was not often detected and described [48]. Recently, the potential for downregulation of the expression of cyclin D-1 in CMTs has been described. 

The cyclin E in conjunction with its CDK2 kinase, plays an important role in inducing the S-phase, DNA-replication, G0 induction and oncogenic transformation [49]. It has been suggested that deregulation of the cyclin E/CDK2 complex causes replication stress in S phase and chromosome segregation errors in M phase which leads to genomic instability and carcinogenesis [50]. Cyclin E is suggested as prognostic factor in HBCs. The correlation between the level of cyclin E and five-year overall survival has been noted [51]. It is in line with other study in which cytoplasmic cyclin E expression correlated with poor prognosis [52]. So far, both cyclin D1 and E are used only as cellular markers.

#### 3.1.2. Proliferation Markers

One of the widely used cell proliferation markers is antigen Ki-67 (Ki-67). In HBC development, the expression of Ki-67 is positively correlated with cancer progression. It can be also an indicator of prognoses and outcomes [53,54]. In addition, it has been proved that the Ki-67 expression in lymph node metastases correlates positively with its expression in the tumor tissues [55,56]. 

Cell nuclear antigen (PCNA) is another biomarker that is widely used in both gynecological and andrological health issues [57,58,59,60,61]. An increased PCNA index in normal mammary glands surrounded by glands with aggressive tumors has been noted [58]. PCNA is also associated with shorter survival in early HBCs [59]. In CMTs it was considered a significant prognostic factor for carcinosarcomas [60]. Ki-67 and PCNA can be used as both: cellular and serum biomarkers.

Human epidermal growth factor receptor 2 (HER-2, also known as HER-2 *neu* or ErbB-2) is also a noteworthy tumor marker [62,63,64,65,66,67]. It is a protooncogene that encodes a glycoprotein responsible for the stimulation of cell proliferation and differentiation. During carcinogenesis, it regulates tumor development and differentiation. It is demonstrated in about 30% of CMT and similarly in HBC. 

In women, HER-2 overexpression is strongly connected with high tumor aggressiveness, lack of response to cytotoxic and endocrine therapies, and in general decreased survival [62]. However, if HER-2 is overexpressed at the early stages of the disease, it can be used as a perfect therapeutic target. Trastuzumab, which is a targeted drug is usually combined with chemotherapy in the early phase of HBC treatment. Similarly like in humans in dogs HER-2 is thought to be a marker of short patient survival. However, not all the studies agree with that statement [64]. It is suggested that HER-2 plays a role in tumor proliferation in canines, but not necessarily in malignant transformation [62]. Various studies confirmed the correlation between HER-2 expression and tumor mitotic index, high histological grade, and size [62,65]. Due to contradictory results in CMT, a recent study was made on quantitative analysis of HER-2 expression by RNA in situ hybridization showing more reliable test results in comparison to immunohistochemistry analysis [67].

Due to confirmed similarity between human and canine HER-2 antigens, immunotherapy with trastuzumab or cetuximab could effectively be used in CMT with HER-2 expression [66]. Still more research in this field is needed.

#### 3.1.3. Apoptosis Markers

The biochemical and morphological changes which occur during the apoptosis in the cell are caused by intracellular cysteine proteases (caspases) [68,69,70]. In mammals, there are two principal apoptotic pathways: death receptor-mediated pathway and mitochondrial-mediated pathway [71] and also less well-known pathways such as the initiator role of caspase-12 or caspase-2 activated by endoplasmic reticulum stress or perforin/granzyme pathway [72,73]. The correlation between caspases and both HBC and CTM is the aim of the studies. Pu et al. showed that high caspase 3 expression is related to worse prognosis in humans [74]. In veterinary medicine, it is suggested that during tumor progression the caspase-dependent apoptotic pathways can be inhibited [75].

Programmed cell death protein-1 (PD-1) is an oncogene that due to its similarity in structure and functions is classified in the CD28 family [69,70]. PD-1 by binding with its ligands (PD-L1 and PD-L2) inhibits T-cell proliferation, cytokine production, and cytolytic function [76,77,78,79,80,81]. It was documented that PD-L1 expression was detected in 19% of triple-negative HBCs [79]. Maekawa et al. found that 80% of the canine mammary tumors showed expression of the PD-L1, however, the correlation with prognosis remains unclear [81]. PD-L1 expression was also shown in a high number of canine malignant melanoma specimens [81]. In addition, the action of PD-1 with its ligands causes the intensification of negative signals sent to the lymphocytes infiltrating the tumor [75]. Due to these facts, the PD-1/PD-L1 blockade seems to be very favorable in immunotherapy. Current research has demonstrated that PD-1 and PDL-1 have a high potential as biomarkers in canine tumors [82].

In both, humans and animals, the P53 is the main tumor suppressor gene [83]. Physiologically, the level of p53 is low or below the detection limit. Activated p53 may influence apoptosis or cell cycle arrest which prevents cancer occurrence in the organism [84,85]. It was suggested that p53 can be a useful biomarker in patients suffering from triple-negative breast cancer (TNBC). It has been proved that high expression of p53 has a positive correlation with poor prognosis [86]. In veterinary medicine, the correlation between p53 high expression and the occurrence of cancers was also considered. Higher p53 expression was noted in large-breed dogs, which could suggest some breed predispositions [87]. However, it was suggested that changes in p53 gene expression may not be correlated with histologic aggressiveness in CMTs [88]. In another study, the p53 expression was connected with high proliferative activity and high histological grade. Due to this fact, Brunetti et al. suggested that p53, ER, and Ki67 can be used to predict the biological behavior of CMT, and they have showed that p53 is associated with higher proliferative activity and higher histological grade of CMTs [89].

### 3.2. Metastatic Potential and Prognosis of the Tumor

Another important group of biomarkers is those that indicate the metastatic potential of the tumor. The ability of the tumor to metastasize depends on the cell’s adhesions to each other or adjacent tissues. The strength of those connections is measured by the levels of expression of the proteins involved in these processes. There are various types of adhesion molecules, such as integrins, selectins, immunoglobulin-like particles, and cadherins. 

#### 3.2.1. Cadherins 

Cadherins are a group of transmembrane glycoproteins that are responsible for cell-cell adhesions. Their main function is to keep the physiological structure of a tissue. However, dysfunction of these adhesions may also play a role during carcinogenesis [90]. There are many cadherins, but classical cadherins are considered epithelial-, placental- and neural-cadherin. The most commonly examined cadherin is epithelial-cadherin (E-cadherin), which takes part in epithelial cellular adhesion [91]. Structurally, cadherins are composed of an extracellular domain (cytoplasmic tail) and a transmembrane domain. Using their cytoplasmic tail they bind to other proteins, known as catenins, and together they form complexes. The common types of catenins are α, β, and p-120. The most common catenin is β-catenin, which is a cytoplasmic protein and disruption of its degeneration during carcinogenesis leads to its accumulation in the cytoplasm and nucleus of tumor cells.

In human medicine, lower expression of E-cadherin corresponds to a higher tumor histological grade, tumor size, and lymph node status, and indicates poor disease outcome. The loss of E-cadherin expression is a signal of epithelial-mesenchymal transition (EMT), a phenomenon occurring during neoplastic progression and is related to increased cell migration and invasion [92]. In CMTs, similarly, downregulation of E-cadherin expression is connected with increased tumor growth and disease spreading, tumor malignancy, the aggressiveness of metastases, and short life expectancy [91,93]. A recent study showed that expression of E-cadherin as well as expression of α, β, and p-120 catenin, remain unchanged in 80% of benign neoplasia cases, but at the same time, their expression in 20% of canine malignant mammary tumors was reduced [94]. The authors also found a correspondence between E-cadherin and p-120 catenin expression as well as a notable correlation between the histological type and the expression of α-catenin in malignant CMTs.

Generally, low expression of E-cadherin (estimated by IHC in tumor samples) is a marker of a poor outcome, however, more studies are needed. 

#### 3.2.2. CEA

Carcinoembryonic antigen (CEA) is a glycoprotein involved in intracellular adhesion. It is located in epithelial cell membranes, and its levels may be increased in case of colon, breast, and lung cancers. In human medicine, CEA is still considered one of the most reliable cancer markers [95]. Especially when evaluated in combination with cancer antigen 15-3 (CA 15-3) they are considered to have good sensitivity and specificity. It has been shown that CEA blood levels have a relationship with the feedback to therapy as well as with the presence of HBCs metastases [96]. An increase in CEA levels may also indicate early recurrence of the disease and metastasis [97,98,99,100]. Moreover, CEA overexpression is related to clinicopathological features of the neoplasia, such as tumor size, tumor grade, and lymph node status [101].

Recently in dogs, a few studies determined CEA levels in canine tissues [95,101,102,103]. In those animals, like in humans, serum levels of CEA are detected in both dogs with CMTs and those that are healthy. However, CEA serum values in dogs are approximately ten times smaller [101]. CEA is measured in both serum and tissue samples using various diagnostic techniques. A recent study showed that CEA serum levels were relatively higher in the case of bitches with carcinoma than in healthy ones [103]. In addition, CEA levels in these bitches also positively correlated with the presence of metastases, tumor dimension, and histological grade. The levels of CEA after mastectomy decreased greatly, suggesting that CEA could be a good biomarker of early relapses and metastases. 

Determination of CEA is only credible when evaluated together with CA 15-3 and may be convenient in an early cancer diagnosis in both humans and dogs. The applications of CEA seem to be multiple as it is also useful for the estimations for patients’ follow-up, which may indicate recurrences and/or metastases. Nevertheless, this field requires more research.

#### 3.2.3. CA 15-3

Cancer antigen 15-3 (CA 15-3, also called Mucin 1 or MUC 1) is also a transmembrane glycoprotein, a product of the *mucin 1* gene [104,105]. What is interesting, during carcinogenesis MUC1 may act as an anti-adhesive molecule and allows the detachment of malignant cells. Therefore, it escalates the possibility of metastases and tissue invasion.

The CA 15-3 is considered a reliable biomarker for controlling treatment effects and confirming relapses and metastases of HBCs. It has a positive correlation with certain neoplastic characteristics, such as lymph node status, tumor dimension, and stage of the illness [98]. There are only a few studies concerning CA 15-3 determination in canine mammary cancers done by three researcher groups, [95,101,102,104,105]. CA 15-3 was found to be expressed in healthy mammary gland tissues, of the bitch as well as in CMTs tissues. The researchers showed that CA 15-3 had a positive correlation with tumor grade. Serum concentrations of CA 15-3 were higher in grade II and III carcinomas than in grade I carcinomas [105]. In one study the researchers even propose average serum values of both CA 15-3 and CEA in bitches with mammary tumors [102]. At the same time, overexpression of CA 15-3 was associated with an unfavorable outcome and a worse prognosis. However, other features of neoplastic disease such as tumor size, skin ulceration, necrosis, inflammation, and histological type of CMTs were rather not associated with the serum CA 15–3 levels. 

As mentioned previously, it is advised to determine CA 15-3 with other biomarkers at the same time, such as CEA to increase its diagnostic sensitivity and specificity. CA 15-3 seems to be a good marker of the patient’s follow-up and diagnosis of recurrences. In dogs, it is a promising biomarker of canine mammary tumors, but more studies are needed.

### 3.3. Hormone Receptors

CMTs hormone receptors are without a doubt the most studied markers [6,24,55,56,57,58,59,75,88]. In bitches, sexual hormones (estrogen and progesterone) play a role in the growth and development of many tissues, including the reproductive tract and mammary gland. Both hormones participate also in carcinogenesis. It was shown that most CMTs express ER (estrogen receptor) and/or PR (progesterone receptor) [106]. Other hormones such as prolactin or oxytocin were found to be involved in the process of tumor growth [107,108,109].

In women, although ER and PR are one of the oldest known biomarkers, there are still one of the most reliable predictive and prognostic markers of HBCs [110,111]. HBCs that are estrogen receptor-positive (ER+) and progesterone receptor-positive (PR+) are considered better-differentiated tumors with more a favorable prognosis, while estrogen receptor-negative (ER-) and progesterone receptor-negative (PR-) HBC are usually related to more aggressive tumor and worse outcomes [111]. ER+ and PR+ tumors can be treated with endocrine-targeted medicaments, like tamoxifen, which enhances life expectancy in most women [112]. 

In bitches like in women, many studies have proved that the expression of ER or PR was more common in non-malignant tumors, and was usually connected to better clinical outcomes [113,114,115]. A recent study showed also that bitches with high serum levels of estrogens and/or with ER+ tumors had a longer timespan without metastases when spayed during mastectomy [116]. Surprisingly, in one study, increased levels of serum estrogens, progesterone, and testosterone were correlated with tumor relapse and/or distant metastases and shorter disease-free periods [107]. Also, in another study ER- and PR+ tumors were linked to a worse prognosis than in the case of ER+ and PR+ tumors, with ER- and PR- tumors having the worst predictive factor of them all [106]. A novel marker, progesterone receptor membrane component 1 (PGRMC1) was studied in canine adenomas and carcinomas as well as in healthy canine mammary gland tissues [117]. The authors revealed that its expression is elevated in healthy mammary gland tissues and in benign tumors in contrast to expression in malignant tissues. When it comes to oxytocin, a recent study showed that benign tumors have higher expression of oxytocin receptors (OTR) than malignant tumors [109]. It may therefore indicate that OTR could be a good marker of cellular differentiation. 

In both HBC and CMT, ER and PR remain very good prognostic markers and their determination is recommended for choosing treatment options. Therapies with anti-estrogen drugs (tamoxifen) are recommended in the case of HBCs in women, but as well may be applied in CMT treatment [118]. However, there is an excessive possibility of pyometra, so ovariohysterectomy is recommended. The application of tamoxifen in CMT and its clinical effects are still under study. 

### 3.4. “Metabolomic” Markers

Metabolomics is defined as a comprehensive and systematic identification and quantification of low molecular weight metabolites (<1500 Da) in biological samples, and the metabolome is a set of all metabolites present in cell, tissue, organ or organism [119,120]. Since the field of metabolomics has been developed, the number of studies related to tumors increased [121]. The changes in metabolic pathways are often connected with cancer progression. It was documented that the urinary biomarkers such as concentrations of 5-hydroxymethyl-2-deoxyuridine and 8-hydroxy-2-deoxyguanosine were increased in HBCs [122]. A comprehensive metabolic map of HBCs has been developed. The cytidine-5-monophosphate/pentadecanoic acid metabolic ratio was found as a differentiator between neoplastic and healthy tissues [123]. 

In veterinary medicine, metabolomic research is still limited and only a few studies focused on different cancers have been performed. In canine bladder cancer, the NMS-based metabolomic study connected with profiling and comparing the urine metabolites was performed. The urea, choline, methyl guanidine, citrate, acetone and β-hydroxybutyrate were identified as highly sensitive biomarker candidates [124]. In a very recent study, higher concentration of three tryptophan metabolites, 5-hydroxyindolacetic acid, serotonin, indoxyl sulphate, and kynurenic acid, two tyrosine metabolites, 3,4-dihydroxy-L-phenylalanine and epinephrine were detected in samples of the urine in CMTs patients [125]. Thus, metabolites of tyrosine and tryptophan are suggested as useful biomarkers in CMTs [125].

Metabolomics seems to be useful for evaluating the cancer prognosis, especially in humans. Higher glycine tissues concentrations indicated poor prognosis in HBCs patients [126]. In the same study lower taurine and glycine tissue concentrations were found in patients who had metastasis and recurrences compared to healthy subjects five years after surgery. Thus, generally in medicine, metabolomics gives the impression of being an effective way for an early diagnosis of cancers and survival rate prognosis. However more research is still needed. 

### 3.5. Gene Expression 

In human breast cancer, gene expression analysis is one of the most valuable diagnostic tools for the identification of novel biomarkers which are predictive and prognostic factors [127]. In addition, it allows for the completion of the histopathological classification of mammary tumors. In women, the current prognostic classification for breast cancer includes 5 phenotypes according to intrinsic molecular subtypes (Luminal A, Luminal B, HER2-enriched, Basal-like, and Claudin-low) [128]. In addition, the study of the gene expression of cancer is a path to the era of personalized medicine which allows precise treatment evaluation [129]. Similar evaluation will highly improve canine mammary tumor management, especially taking into account the complexity of this kind of tumor in dogs.

Global gene expression profiling is a future direction in veterinary medicine. In canine mammary carcinomas of different grades (I, II, and III) five genes (SEHRL, ZFP37, MIPEP, RELAXIN, and MAGI3) that had an influence on tumor biologic behavior were identified [130]. In a recent study, in which 1.699 genes were differentially expressed, the overexpression of the HYAL-1 gene was recognized as a possible biomarker linked with cell growth, migration, invasion, and angiogenesis [131]. Also, genes, such as COL11A1, SFRP2, LCN2, COL2A1, and H19. COL11A1, MMP3, MMP1, AREG, PTHLH, and SFRP2, influence the cell cycle, apoptosis, dendritic cell maturation, DNA recombination and repair, Wnt/β-catenin signaling, migration, and angiogenesis were altered in benign and malignant CMTs [132]. In addition, recent studies suggest that canine BRCA2 which is involved in DNA repair seems to be a very important marker as well as in humans [133,134,135,136]. The 97.9% of dogs with CMT had one up to three genetic variations out of the seven thus, low expression of BRCA2 plays a part in mammary tumor growth. Also, several single nucleotide polymorphisms in RAD51 (rs23623251 and rs23642734) and one in the STK11 gene (rs22928814) may be associated with the risk of CMT [137]. However, limited information is available regarding precise mutations and the mechanism of them in mammary neoplasms. More research on the canine genome and particular mutations is needed. However, some gene mutations such as BRCA2 may allow for prediction of the predisposition for cancer development in the future in dogs.

### 3.6. miRNA

The novel biomarkers in human oncology are small noncoding RNA molecules called microRNAs (miRNA). miRNAs make particularly good biomarkers because they can be secreted in biofluids such as serum, urine, and breast milk [138]. Thus, the collection of samples is of low invasiveness. In women with breast cancer, several miRNAs seem to have a diagnostic and prognostic potential (miR-9, miR-10b, mir-17-5p, mi-R-148a, and miR-335) [139]. In human breast cancer, one of the most promising biomarkers is miR-21 which promotes cell proliferation and metastasis [140]. However, in dogs, the data is missing. It was documented, the same as in humans, that in dogs miRNAs such as cfa-miR-96 and-149 are oncogenic and tumor-suppressive [141]. However, more research is needed because in earlier mentioned study a distinct regulatory mechanism of cfa-miR-8832 expression in both species was discovered. In addition, the level of miRNA expression may indicate the metastatic potential and prognosis for tumor patients [140,142]. Recently, sixty-five individual miRNAs were performed in canine mammary carcinoma [138]. In this study, serum miR-19b was evaluated as a candidate for diagnostic purposes, whereas miR-18a seemed to be a good prognostic factor. Also, miR-214 and miR-126 were remarkably up-regulated in the serum of bitches with mammary carcinoma [142]. However, the miRNA expression may differ in tumor tissue and blood. In one study, the cfa-miR-144, cfa-miR-32 and cfa-miR-374a levels in blood were different from those in tumor tissues [142]. miR-21 seems to be a very promising serum biomarker because it appears to be more sensitive than other commonly used markers such as CA 15-3 (12.84-fold vs. threefold) [143].

Thus, miRNAs are very promising markers for cancer diagnosis. Multiple miRNA-based profiles may also allow for predicting prognosis and the metastatic potential of mammary tumors. In addition, because of its regulatory function, miRNA-based therapy seems to be very promising. However, there is still a lack of studies regarding miRNA analysis in dogs.

### 3.7. Transcriptome Sequencing

Recently, many studies of CMT are focused on transcriptome analysis in order to find specific genes that up-regulate various biological processes associated with tumorigenesis. In one study, 351 differentially expressed genes (DEGs) were identified in CMT [144]. Eight of those DEGs were previously stated as cancer-associated genes in HBC. In another study, whole-exome sequencing (WES) and whole-transcriptome sequencing (WTS) of CMT were performed in order to discover single nucleotide variations and mutations [145]. In a recent study, similarities in terms of core oncogenic signatures containing principal genes of the PI3K-Akt and p53 pathways were recognized [146]. In addition, it was summarized that the relative paucity of aneuploidy and SCNA drivers, such as *ERBB2* amplifications in CMTs, is a late-stage marker in humans. The obtained extensive data can be used in comparative oncological studies between canine and women breast cancers especially taking into account that the neoplastic process in bitches takes relatively less time than in women.

### 3.8. Inflammatory Markers

#### 3.8.1. Inflammatory Cells Infiltration

Tumors are generally infiltrated by inflammatory cells which interfere with cell survival, migration, and invasion of human breast cancer. Tumor-infiltrating macrophages (TIMs) are one of the most abundant immune cells in the tumor immune-infiltrating microenvironment. In humans, a higher density of TIMs is associated with worse clinical course in some tumor types and with poor prognosis in others [147]. In canine mammary gland tumors, characteristics of immune cells also seem to be a good prognostic factor [148]. In a very recent study, the clinical significance of the specific histologic locations of macrophages, such as (iTIM), stromal (sTIM), and total count (tTIM) was documented [149]. The more aggressive tumor features were related to a higher number of tTIM, whereas a high number of sTIM and iTIM were connected with nodal metastasis. In dogs with III grade CMTs, the number of CD204-positive macrophages was greater than in grades I and II [150]. It suggests that CD204-positive macrophages can have an influence on the growth and course of CMTs.

In addition, lymphocyte infiltration is also associated with predicting clinical outcomes. In HBCs, T cells are strictly involved in tumor advancement. Some of them, especially T regulatory cells (Tregs), may have immunosuppressive properties which in special circumstances may be beneficial for tumor immunity. In CMTs shorter survival time was documented in patients with high CD3+ and CD4+ cells in the tumor tissue [151]. Also, a large number of intratumoral FoxP3+ cells was related to defective differentiation of tumors, high histological grade of malignancy, and increased angiogenesis thus guarded prognosis [151]. In a very recent study, the number of Foxp3+ cells and CCR4+, which are associated with Tregs, was increased in the tumor tissues of mammary carcinoma in comparison to the healthy tissues [152]. This finding suggests that Treg recruitment may be influenced by CCL17/CCR4 axis. Thus, its’ blockade may also be a future choice for tumor therapy by decreasing the Treg number. Lymphocyte populations may be measured also in peripheral blood. In one study, in CMTs stage II or III, the white blood cell count and neutrophils were significantly increased, whereas the absolute number of lymphocytes remain unchanged [153]. However, CD8+ cells decreased leading to the increase of the CD4+/CD8+ ratio suggesting that cell-mediated immunity is altered in bitches with CMTs.

Canine mammary cancer stem-like cells are identified as CD44+/CD24- phenotype cells. Higher number of those cells is connected with high grade and lymph node infiltration [154]. In addition, it was suggested that CD24 expression seems to be a profitable biomarker of bad prognosis.

#### 3.8.2. Other Inflammatory Tissue and Blood CMTs Markers

In both, human and canine studies cyclo-oxygenase-2 (COX-2) overexpression in mammary tumors is one of the hallmarks of cancer and disease progression [155,156]. It is documented that COX-2 expression influences mammary tumorigenesis by inducing angiogenesis and cell proliferation, promoting metastasis and tumor-associated inflammation. In one study, high expression of COX-2 was correlated with CD31, vascular endothelial growth factor (VEGF), Ki-67, CD3+ T-lymphocytes infiltration, MAC387 macrophages number or synuclein gamma, tribbles 1 which suggests the worst prognosis [157,158]. In CMT tissues, increased expression of the third isoform of nitric oxide (iNOS), COX-2, and VEGF were related to the histological grade of malignancy [159].

Versican is an extracellular matrix proteoglycan that causes the TIMs production of pro-inflammatory and pro-tumoral cytokines and chemokines, such as tumor necrosis factor-α (TNF-α), tumor growth factor-β1 (TGF-β1), vascular endothelial growth factor (VEGF), and CC chemokine ligand 2 (CCL2) [160]. It was documented that versican expression is also correlated with accumulation in mammary carcinoma tumors. CMT cytokines expression seems to be a marker of canine mammary tumors malignancy. A recent study showed that rivoceranib (a novel tyrosine kinase inhibitor) had an anti-proliferative effect on tumors expressing vascular endothelial growth factor receptor-2 (VEGFR2) [161]. In one study, 60 pro-inflammatory cytokines were evaluated [162]. In this study, CMTs showed higher expression of CKβ-8-1, EGF, NAP-2, and PARC than healthy mammary tissue. The higher expression of MCP-1, MCP-2, PDGF-BB, RANTES, and SCF was documented in tumors with higher aggressiveness. Also, tissue overexpression of IL-35 was connected with worse overall survival due to increased histological grade of malignancy, mitotic index, neoplastic intravascular emboli, and lymph node metastasis [163] in CMTs patients. Also, cytokines connected with T lymphocyte activity are upregulated CXCR3, CCR2, IL-4, IL-12p40, and IL-17 in CMTs [164]. However, the blood IL-17 was unchanged.

Specific proteins may be measured in the blood which is less invasive and sometimes more sensitive than CMTs biopsy. The immune system responds to low, generally undetectable levels of tumor-associated autoantigens which are potential biomarkers of faster canine mammary tumor identification. In a very recent study, 5 canine mammary tumor-associated autoantigens (TPI, PGAM1, MNSOD, CMYC & MUC1) were evaluated [165]. In this study, among malignant CMTs, autoantibodies were present across all tumor grades. However, the MUC1 antibodies were related to 80% of grade III, thus indicating higher aggressiveness of the tumor. Several heat shock proteins (HSPs) are involved in the recognition and prognosis of specific neoplasias [166]. As they protect the cancer cells from apoptosis, HSPs are suggested as promising biomarkers of carcinogenesis. It is documented that HSP27 is greatly raised in tumor subjects in comparison to healthy controls [167].

Acute phase proteins (APPs) are unspecific markers of inflammation which are altered by any kind of disturbances in homeostasis. The main APPs in dogs is C-reactive protein (CRP) [168,169,170]. Serum CRP concentration was significantly higher in bitches with malignant and metastatic tumors, in contrast to bitches with benign tumors and control [169]. However, it is not good enough marker for small tumors less than 3 cm. Thus, it suggests that CRP probably may be used only in advance stages. Not to forget, other pathological conditions, such as pyometra, induce an increase in its serum concentration. Therefore, this biomarker is not very specific.

Recently, many studies attempted to categorize mammary gland tumors into subtypes in order to choose the proper individual approach [113,132,169]. Molecular phenotyping of tumors on the basis of various cellular biomarkers expression (PR, ER, HER-2, EGFR, Ki-67) has identified luminal A (ER^+^ and/or PR^+^, HER2^-^), luminal B (ER^+^ and/or PR^+^, HER2^+^), triple-negative basal-like/non-basal like (ER^-^ and/or PR^-^, HER2^-^), and HER_2 overexpressing CMT (lack of ER and PR expression) subtypes. Each molecular tumor subtype corresponds with a different histological type, grade, tumor aggressiveness, and prognosis. This helps to standardize treatment options and to estimate prognosis, but it is still not very precise and further studies are needed.

## 4. Conclusions

The high risk of developing CMT poses a serious threat to the life and health of the bitches and should always be considered when treating older bitches. In the case of HBC, there are many diagnostic and treatment protocols that greatly decreased the rate of death among patients in recent years. In dogs, however, while the nature of mammary gland neoplasia is similar, CMTs are still poorly studied, usually detected at advanced stages, lacking in treatment options and have a poor prognosis. Therefore, an early diagnosis of CMTs and the introduction of personalized treatment should be considered. In this article, we summarized possible novel diagnostic techniques as well as main biomarkers that seem to improve treating CMTs. The new cytological grading system of CMTs or B-mode ultrasound, Doppler technique, contrast-enhanced ultrasound, and real-time elastography may be useful in pre-surgical evaluation. However, the sensitivity and specificity of these techniques increased when they are evaluated together with some serum and tissue biomarkers. Especially, in order to detect cancer before macroscopic changes are visible. We described the main biomarker groups, among them markers of the cell cycle, proliferation, apoptosis, metastatic potential and prognosis, hormone receptors, inflammatory, and more recent: metabolomic, gene expression, miRNA, and transcriptome sequencing markers. We summarized the list of biomarkers and their uses in a table (Table 1). Ki-67, HER-2, E-cadherin, ER, and COX-2 are well-known tissue biomarkers, and their evaluation should always be considered in histopathological samples. The use of more recent CMTs markers such as CEA and CA 15-3 as well as multiple miRNAs, transcriptome sequencing, and some gene expression mutations is considered profitable. Also, the PD-1/PD-L1 blockade gives an impression of being very auspicious in immunotherapy, which can be an effective treatment of CMTs. The metabolomic markers though poorly studied may be beneficial for an early tumor diagnosis. The use of several of the above-mentioned markers together seems to be the most effective way for obtaining an early diagnosis of neoplastic disease as well as to evaluate response to treatment, presence of tumor progression, or further prognosis.

Nevertheless, further research in this field is needed. There is a huge need for standardized clinical study among dogs with CMT in order to improve the clinical approach to diagnosis CMT and applying targeted therapies. As mentioned before, according to one health concept investigation in both veterinary medicine and human medicine have mutual benefits [171,172].

## Figures and Tables

**Figure 1 vetsci-09-00526-f001:**
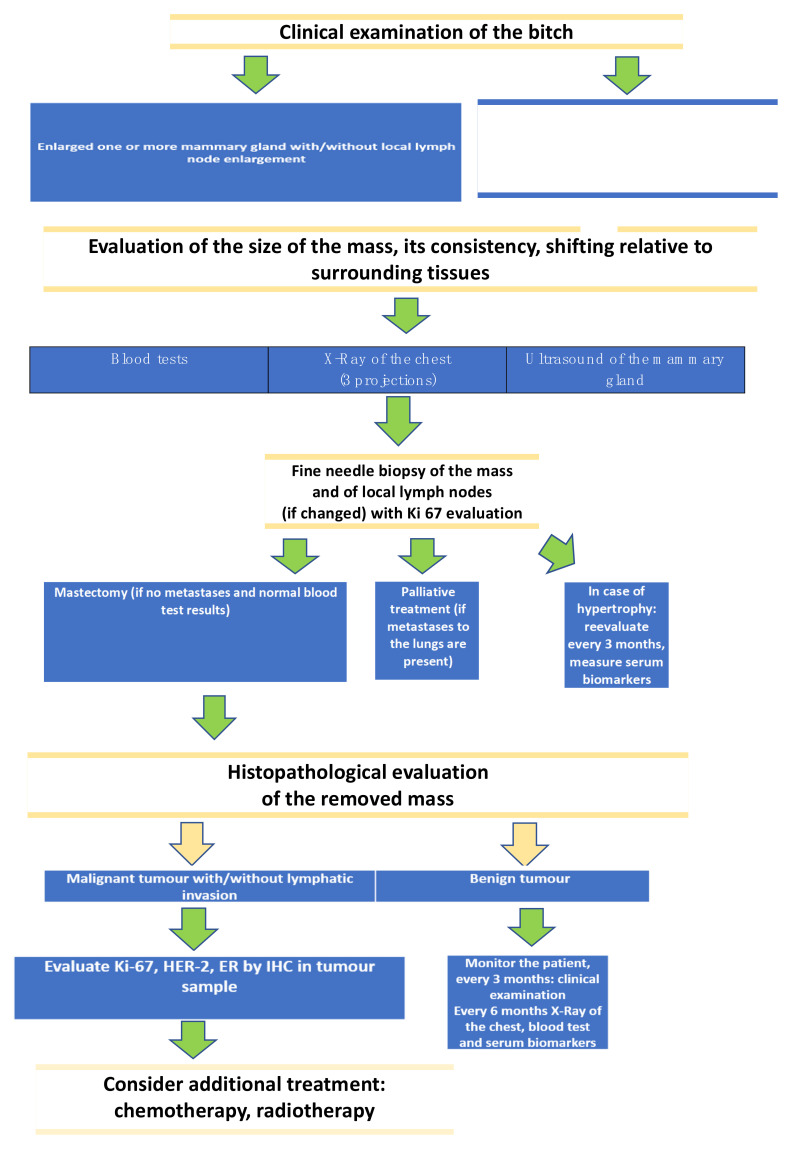
Proposal of the procedure of canine mammary tumors diagnostics.

**Table 1 vetsci-09-00526-t001:** Tissue and fluid biomarkers of mammary gland tumors.

Biomarker/Biomarkers	Function	Main Finding	References
cycylin D1, cyclin E1,	Cell cycle	cyclin D1 overexpression is commonly related with metastasis and shorter life expectancy	[43,44,45,46,47,48,49,50,51,52]
Ki-67, PCNA	Proliferation	expression of Ki-67 and PCNA positively correlates with cancer progression	[53,54,55,56,57,58,59,60]
HER-2	Proliferation	there is a correlation between HER-2 expression and tumor mitotic index, high histological grade, and tumor size	[62,63,64,65,66,67]
PD-1	Apoptosis	most canine mammary tumor show expression of the PD-L1	[68,69,70,71,72,73,74,75,76,77,78,79,80,81,82]
p53	Apoptosis	p53 expression was connected with high proliferative activity and high histological grade	[83,84,85,86,87,88,89,90]
E-cadherin	Cell adhesion	downregulation of E-cadherin expression is connected with increased tumor growth and disease spreading, tumor malignancy, the aggressiveness of metastases, and short life expectancy	[91,92,93,94,95]
CEA, CA 15-3	Cell adhesion	both positively correlate with the presence of metastases, tumor dimension, and histological grade, it is recommended for them to be evaluated together	[96,97,98,99,100,101,102,103,104,105,106]
ER, PR	Hormone receptors	the expression of ER or PR was more common in non-malignant tumors, and usually was connected with better clinical outcome	[107,108,109,110,111,112,113,114,115,116,117,118,119]
5-hydroxyindolacetic acid, serotonin, indoxyl sulphate, and kynurenic acid, 3,4-dihydroxy-L-phenylalanine and epinephrine	Metabolites of tyrosine and tryptophan	detected in samples of the urine in CMTs patients	[125,126]
HYAL-1	Gene encoding lysosomal hyaluronidase	a possible biomarker linked with cell growth, migration, invasion, and angiogenesis	[132]
BRCA1, BRCA2	Genes involved in DNA repair	The 97.9% of dogs with CMT had one up to three genetic variations out of the seven	[134,135,136,137]
miR-214 and miR-126	miRNA	remarkably up-regulated in serum of bitches with mammary carcinoma	[143]
miR-18a	miRNA	seemed to be a good prognostic factor	[139]
miR-21	miRNA	be more sensitive than other commonly used markers	[144]
CD204+ macrophages	Inflammatory cell infiltration	In dogs with III grade CMTs, the number of CD204-positive macrophages was greater than in grades I and II	[151]
LT CD3+, LTCD4+	Inflammatory cell infiltration	high CD3+ and CD4+ cells in the tumor tissue correlate with shorter survival time	[152]
LT FoxP3+	Inflammatory cell infiltration	a big number of intratumoral FoxP3+ cells was related to defective differentiation of tumors, high histological grade of malignancy, and increased angiogenesis, thus guarded prognosis	[153]
COX-2	Inflammatory markers	increased expression is correlated with the histological grade of malignancy	[161]
VEGF, TNF-α, TGF-β1	pro-inflammatory and pro-tumoral cytokines and chemokines	CMT cytokines expression seems to be a marker of canine mammary tumors malignancy	[162]
MCP-1, MCP-2, PDGF-BB, RANTES, SCF	pro-inflammatory cytokines	expressed in tumors with higher aggressiveness	[164]
IL-35	interleukin	connected with worse overall survival by increased histological grade of malignancy, mitotic index, neoplastic intravascular emboli, and lymph node metastasis	[165]
CXCR3, CCR2, IL-4, IL-12p40	pro-inflammatory cytokines and interleukines	cytokines connected with T lymphocyte activity are upregulated	[166]

## Data Availability

Not applicable.

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
