# Peer review of "The Novel Diagnostic Techniques and Biomarkers of Canine Mammary Tumors"

_vetsci, 2022, doi:10.3390/vetsci9100526_

Round 1
Reviewer 1 Report (Previous Reviewer 4)
The content of the manuscript is very interesting, but the text still needs English revision, especially the introduction chapter. For example, there are a lot of commas and articles lacking in the text, which greatly impairs its reading. Please, submit the manuscript to an English reviewer before resubmitting for publication. Sometimes, it looks like it has been written by different people, since the quality of writing varies throughout the text. Some examples are showed below.
L58. Do the authors mean “ethology” or “etiology”?
L94. Remove the “s” before “is usually”.
L123. It should be “exists” instead of “exist”. In fact, the whole sentence should be rewritten: In the actual histological staging system, a specific category for CMTs mammary carcinomas in situ exists. On the other hand, are you sure you want to say “actual” or “current”?
L160. Remove the comma after “Vannozi et al.”.
L170. It should be “In a recent study”.
L290. “Has to do” is not a scientific term. The authors could use “to be related” for example.
And the list goes on. So, please, submit the manuscript to an English reviewer.
OTHER COMMENTS
When talking about cyclins E (L215), this reviewer considers more appropriate to talk about cyclin (and not cyclins) E presenting two different types, E1 and E2.
L409-410. Write anti-estrogen drugs instead of anti-estrongen drugs.
Regarding to the proposed algorithm for CMTs diagnosis, authors seem to recommend a “wait and see” approach when tumours are small. I strongly disagree with this approach. If the intention of the algorithm is a different one, please clarify it so it can be easier to interpret.
Author Response
Dear reviewer 1! Thank you for your kind revision. We tried to answer all your comments and we hope that in the present version the manuscript is suitable for publication.
The content of the manuscript is very interesting, but the text still needs English revision, especially the introduction chapter. For example, there are a lot of commas and articles lacking in the text, which greatly impairs its reading. Please, submit the manuscript to an English reviewer before resubmitting for publication. Sometimes, it looks like it has been written by different people, since the quality of writing varies throughout the text. Some examples are showed below.
Thank you for that comment, The manuscript has been revised, especially the introduction part and additional comas and articles were put.
L58. Do the authors mean “ethology” or “etiology”?
We meant etiology, sorry for the error, it was due to Microsoft office auto-corrections.
L94. Remove the “s” before “is usually”.
We corrected that mistake.
L123. It should be “exists” instead of “exist”. In fact, the whole sentence should be rewritten: In the actual histological staging system, a specific category for CMTs mammary carcinomas in situ exists. On the other hand, are you sure you want to say “actual” or “current”?
Thank you. The sentence has been rewritten.
L160. Remove the comma after “Vannozi et al.”
The coma has been removed.
L170. It should be “In a recent study”.
Thank you. We corrected it.
L290. “Has to do” is not a scientific term. The authors could use “to be related” for example.
Thank you. We corrected it.
And the list goes on. So, please, submit the manuscript to an English reviewer.
Thank you for your suggestion. We will submit the manuscript to a professional english editing service.
OTHER COMMENTS
When talking about cyclins E (L215), this reviewer considers more appropriate to talk about cyclin (and not cyclins) E presenting two different types, E1 and E2.
Thank you. This part has been corrected.
L409-410. Write anti-estrogen drugs instead of anti-estrongen drugs.
Thank you. We corrected it.
Regarding to the proposed algorithm for CMTs diagnosis, authors seem to recommend a “wait and see” approach when tumours are small. I strongly disagree with this approach. If the intention of the algorithm is a different one, please clarify it so it can be easier to interpret.
Thank you for your suggestion. We corrected the algorithm.
Reviewer 2 Report (New Reviewer)
Ilona Kaszak et al., comprehensively reviewed diagnostic techniques and biomarkers for canine mammary tumors, which provide an important foundation for developing new diagnostic methods in the future. However, the organization of contents, wording, and grammar need to be substantially revised. Please see following comments and suggestions:
* Since tittle is "...diagnostic techniques and biomarkers...", it's better to reorganizing the content by dividing the main text into two parts: techniques and markers. For instance, combine section 2 (cytology) and section 3 (ultrasound), which are both about techniques. And combine Section 4, 5, 6,7, 8, 9, 10, and 11 which are all about biomarkers. Then use sub-tittles to describe distinct techniques and biomarkers under two main parts.
* line 58. "ethology" is inappropriate. I guess what you mean is "etiology"?
* line 59. What do you mean "mayor"?
* line 64. "build" should be "built"
* line 74. "valid" should changed to "validated"
* line 113. you don't need to repeat to label the abbreviation "N" for nodal stage
* line 121. "Canadas" is not a correct word
* line 216. Check grammar "..in including.."
* line 243. "HER-2 overexpressed ...is a therapeutic target" is very confusing
* line 392. "reseaches" is not a wrong word but better to be used as a mass noun.
* line 419. The sentence "It is associated with changes..." is hard to understand.
* line 442. No.8-Gene expression and No.10. Tanscriptome sequencing are largely overlapped. Please combine or re-organize these two segments.
* line 501. check the grammar of this sentence.
*line 570. The sentence " Also tissue overexpression of..." is hard to understand.
* line 577. "respond" is a better word than "replies".
* line 635. There is a typo "medine"
* line 637. "algorithm" is not appropriate.
* in addition to suggestions mentioned above, it is highly recommended to get professional english editing service to revise the manuscript.
Author Response
Dear reviewer 2! Thank you for your kind revision. We tried to answer all your comments and we hope that in the present version the manuscript is suitable for publication.
Ilona Kaszak et al., comprehensively reviewed diagnostic techniques and biomarkers for canine mammary tumors, which provide an important foundation for developing new diagnostic methods in the future. However, the organization of contents, wording, and grammar need to be substantially revised. Please see following comments and suggestions:
Thank you for your suggestion.
* Since tittle is "...diagnostic techniques and biomarkers...", it's better to reorganizing the content by dividing the main text into two parts: techniques and markers. For instance, combine section 2 (cytology) and section 3 (ultrasound), which are both about techniques. And combine Section 4, 5, 6,7, 8, 9, 10, and 11 which are all about biomarkers. Then use sub-tittles to describe distinct techniques and biomarkers under two main parts.
Thank you for your suggestion. We tried to reorganize the sections.
* line 58. "ethology" is inappropriate. I guess what you mean is "etiology"?
Thank you. It was an error produced by Microsoft office auto-corrections. We meant etiology.
* line 59. What do you mean "mayor"?
We meant major, sorry for the spelling error.
* line 64. "build" should be "built"
Thank you. We corrected it.
* line 74. "valid" should changed to "validated"
Thank you. We corrected it.
* line 113. you don't need to repeat to label the abbreviation "N" for nodal stage
Thank you. We corrected it.
* line 121. "Canadas" is not a correct word
We meant Ana Canadas and her team
* line 216. Check grammar "..in including.."
Thank you. We meant in inducing not including.
* line 243. "HER-2 overexpressed ...is a therapeutic target" is very confusing
Thank you. We corrected this sentence.
* line 392. "reseaches" is not a wrong word but better to be used as a mass noun.
Thank you. We corrected it.
* line 419. The sentence "It is associated with changes..." is hard to understand.
Thank you. We corrected it.
* line 442. No.8-Gene expression and No.10. Tanscriptome sequencing are largely overlapped. Please combine or re-organize these two segments.
Thank you for that suggestion, be we consider them as two independent sections.
* line 501. check the grammar of this sentence.
Thank you. We corrected it.
*line 570. The sentence " Also tissue overexpression of..." is hard to understand.
Thank you. We corrected the sentence.
* line 577. "respond" is a better word than "replies".
Thank you. We corrected the sentence.
* line 635. There is a typo "medine"
Thank you. We corrected the sentence.
* line 637. "algorithm" is not appropriate.
Thank you. We change the word for procedure.
* in addition to suggestions mentioned above, it is highly recommended to get professional english editing service to revise the manuscript.
Thank you for your suggestion. We will submit the manuscript to a professional english editing service.
Round 2
Reviewer 2 Report (New Reviewer)
No more comments regarding contents from me. However, there are still some spelling or format errors. For instance, both "HER_2" and "HER-2" were found in the text. Please be consistent. Similar problems are found in other gene names as well.
please do thorough proof-reading before the manuscript published.
This manuscript is a resubmission of an earlier submission. The following is a list of the peer review reports and author responses from that submission.
Round 1
Reviewer 1 Report
The work is comprehensive reviewing potential biomarkers in canine CMT.
Perhaps genomics and epigenomic alterations should also be included?
Not sure this statement on page 3 is true: “Mostly β-catenin, which is a cytoplasmic protein, degraded during carcinogenesis, and due to that it is accumulated in the cytoplasm and nucleus of tumor cells.” Is it that disruption of β-catenin degradation process, which leads to its accumulation in the nucleus, leads to tumorigenesis?
Reviewer 2 Report
The manuscript "The novel diagnostic techniques and biomarkers of canine mammary tumours" by Kaszak I. et al., Ref: Manuscript ID: vetsci-1746162, is an excellent review of innovative diagnostic techniques and biomarkers used for the identification and characterization of canine mammary tumors. It is obvious that a meticulous search of the literature has been made, and the outcome has been presented in detail. My comments are really minor, and I suggest including tables and/or figures to summarize and/or highlight the main points of the manuscript.
Reviewer 3 Report
This manuscript aims to review the novel diagnostic techniques and biomarkers of the canine mammary tumor. The authors evaluated Cytology grading, ultrasound-based examination and biomarkers in diagnosis of canine mammary carcinomas. Despite being a very interesting topic in the field of feline medicine, the sections about biomarkers should be expanded as these sections are superficial and lack in-depth explanation of the cited studies. The summary should not just “more studies are needed”. In addition, authors should be focused in selected novel/important/solid biomarkers and explained clearly. I suggested reorganized the Table 1, added the correlation between marker and clinical significance according to the finding of cited studies.
|
Biomerker |
Clinical significance |
Main finding |
Reference |
Reviewer 4 Report
The present manuscript aims to review the utility of biomarkers in the diagnosis of canine mammary tumours. The review is interesting, but it’s not acceptable in its present form
L62-63. This statement was first said by Dr Schmidt in 1969 and not by all the authors named as references. On the other hand, this information is absolutely obsolete and should be removed from the manuscript. If the authors take a careful look to that manuscript, they will realize that it is not valid anymore. It is about time to forget about this really old dogma. I would suggest to read a very nice and interesting presentation performed by Dr Julia Gedon (“Breaking old dogmas: do we have to rethink about canine mammary tumours?”) at the EVSSAR conference of 2018. So, please, rephrase and rewrite this sentence. Update the information.
L383. ER and PR should be defined. For those that have been working for some time, it is clear that they mean estradiol and progesterone receptors. But younger researchers maybe would need that information. In addition, abbreviatures definition is always mandatory. In addition, the second sentence on this line seems incomplete: Other hormones such as prolactin or oxytocin and were. Either there’s information lacking or the “and” has to be removed. Check it, please.
L410-411. Change anti-estrongen by anti-estrogen
L589-594. Be careful with CRP. Other pathological conditions, such as pyometra, induce an increase in its serum concentration.
Another general comment is the fact that, for this reviewer, it is not always clear enough if the biomarkers are analyzed in serum or in mammary tissue. Please, rewrite in some way so it is more understandable.
This reviewer has detected several gramma and orthography mistakes. Since there are so many mistakes, this reviewer has decided not to list them one by one in the present document. Thus, the manuscript needs a deep English revision for both gramma and orthography. For example, commas are lacking or not written in the appropriate location in many sentences, making the reading a little bit confusing and hard sometimes. Also writing style needs to be checked. Some sentences should be rewritten. I suggest to submit to an English reviewer before re-submitting it for publication.